# PRAME Immunoexpression in 275 Cutaneous Melanocytic Lesions: A Double Institutional Experience

**DOI:** 10.3390/diagnostics12092197

**Published:** 2022-09-09

**Authors:** Gerardo Cazzato, Eliano Cascardi, Anna Colagrande, Vincenzo Belsito, Lucia Lospalluti, Caterina Foti, Francesca Arezzo, Miriam Dellino, Nadia Casatta, Carmelo Lupo, Luigi Buongiorno, Alessandra Stellacci, Maricla Marrone, Giuseppe Ingravallo, Eugenio Maiorano, Leonardo Resta

**Affiliations:** 1Section of Pathology, Department of Emergency and Organ Transplantation (DETO), University of Bari “Aldo Moro”, 70124 Bari, Italy; 2Department of Medical Sciences, University of Turin, 10124 Turin, Italy; 3Pathology Unit, FPO-IRCCS Candiolo Cancer Institute, Str. Provinciale 142 Km 3.95, 10060 Candiolo, Italy; 4Pathology Unit, Humanitas Research Hospital and Humanitas University, 20089 Rozzano, Italy; 5Section of Dermatology and Venereology, Department of Biomedical Sciences and Human Oncology (DIMO), University of Bari “Aldo Moro”, 70124 Bari, Italy; 6Section of Gynecology and Obstetrics, Department of Biomedical Sciences and Human Oncology (DIMO), University of Bari “Aldo Moro”, 70124 Bari, Italy; 7Innovation Department, Diapath S.p.A., Via Savoldini n.71, 24057 Martinengo, Italy; 8Section of Legal Medicine, Interdisciplinary Department of Medicine, Bari Policlinico Hospital, University of Bari Aldo Moro, 70124 Bari, Italy

**Keywords:** malignant melanoma, preferentially expressed antigen in melanoma, PRAME, skin, differential diagnosis, dermatopathology

## Abstract

In recent years, the preferentially expressed antigen in melanoma (PRAME) has also been used in the histopathological diagnosis of melanocytic lesions, in order to understand if it could constitute a valid, inexpensive, and useful resource in dermatopathological fields. We performed a double-center study to evaluate whether the data on the usefulness and possible limitations of PRAME could also be confirmed by our group. From 1 December 2021 to 29 March 2022, we collected 275 cases of melanocytic lesions that were immunostained with PRAME (Ab219650) and rabbit monoclonal antibody (Abcam). To better correlate the PRAME expression with its nature (benign, uncertain potential for malignancy, or malignant), we categorized PRAME tumor cells’ percentage positivity and intensity of immunostaining in a cumulative score obtained by adding the quartile of positive tumor cells (0, 1+, 2+, 3+, 4+) to the PRAME expression intensity in tumor cells (0, 1+, 2+, 3+). Of these 275 lesions, 136 were benign, 12 were of uncertain potential for malignancy (MELTUMP or SAMPUS or SPARK nevus), and 127 were malignant. The immunoexpression of PRAME was completely negative in 125/136 benign lesions (91.9%), with only a few positive melanocytes (1+) and intensity 1+ in the remaining 11 cases (8.1%). Of the 127 cases of melanoma (superficial spreading, lentigo maligna, and pagetoid histotypes), PRAME was strongly positive in 104/127 cases (81.8%) with intensity 4+ and 3+. In 17 cases (13.3%; melanoma spindle and nevoid cell histotypes), PRAME was positive in percentage 2+ and with intensity ranging from 2+ to 3+. In 7 cases (5.5%) of desmoplastic melanoma, PRAME was 1+ positive and/or completely negative. Of the 12 cases of lesions with uncertain potential for malignancy, the immunoexpression of PRAME was much more heterogeneous and irregularly distributed throughout the lesion. These data are perfectly in agreement with the current literature, and they demonstrate that the reliability of PRAME is quite high, but its use cannot cause physicians to disregard the morphological information and the execution of other ancillary immunohistochemical stains such as Melan-A, HMB-45, MiTF, and SOX-10.

## 1. Introduction

Historically, the dermatopathological diagnosis of malignant melanoma (MM) has always been the gold standard for obtaining a basis on which to build the therapeutic diagnostic path of patients suffering from this neoplasm [1,2]. Over time, and after the advent of the indispensable and fundamental *Allen–Spitz* criteria, various ancillary immunohistochemical techniques have become more and more widespread, allowing an additional possibly helpful resource in challenging melanocytic lesions [3]. Several markers were used including Melan-A/MART1 [4], human melanoma black-45 (HMB-45) antigen [5], S-100 protein [6], p16 [7], MiTF [8], and, relatively recently, the preferentially expressed antigen in melanoma (PRAME) [9]. In the last few years, more studies have tried to shed light on the reliability of PRAME in the differential diagnosis of atypical pigmented lesions, and it seems that there are different results depending on the authors considered while acknowledging that there is some agreement on some of the results. Furthermore, the interest in PRAME has not focused solely and exclusively on MM but also on other entities such as ocular melanoma and various non-melanocytic malignant neoplasms, including non-small cell lung cancer, breast carcinoma, renal cell carcinoma, ovarian carcinoma, leukemia, synovial sarcoma and myxoid liposarcoma [10,11,12,13]. In this study, we present data from a double institutional experience about the use of PRAME in relation to 275 melanocytic lesions analyzed from 1 December 2021 to 29 March 2022, divided by benign lesions represented by nevi, lesions of non-univocal classification (the so-called MELTUMP, SAMPUS, or SPARK nevus) and frankly malignant lesions, represented by melanoma. Finally, we discuss our findings in light of the evidence in the literature and try to trace future research perspectives related to this topic.

## 2. Materials and Methods

### 2.1. Selection and Classification of Cases

We consulted the archives of two pathological anatomy departments planning to extrapolate their cases from 1 December 2021 to 29 March 2022, on which it was necessary, in addition to immunostaining with other antibodies (Melan-A, HMB-45, MiTF, etc.), to run PRAME to gain a further indication of the benignity/malignancy of the lesions. We identified 323 melanocytic lesions, which were all diagnosed by two dermatopathologists at the time of diagnosis and double-blinded for the execution of this study (A.C and G.C.). After screening, only the cases in which the diagnosis was concordant between both dermatopathologists were considered.

This study was approved by the local ethical committee.

### 2.2. PRAME Immunostaining

First, 5-micron thick tissue sections were cut from formalin-fixed paraffin-embedded (FFPE) blocks. The cases in question were all stained with anti-PRAME antibody Ab219650, rabbit monoclonal, in 1:250 dilution.

To better correlate the PRAME expression with its nature (benign, uncertain potential for malignancy or malignant), we categorized PRAME tumor cells’ percentage positivity and intensity of immunostaining in a cumulative score obtained by adding the quartile of positive tumor cells (0, 1+, 2+, 3+, 4+) to the PRAME expression intensity in tumor cells (0, 1+, 2+, 3+). More specifically, we used the following scores for the percentage positivity of tumor cells: 0% (score 0), 1% to 25% (score 1+), 26% to 50% (score 2+), 51% to 75% (score 3+), and 76% to 100% (score 4+). Furthermore, we used a score for intensity by measuring the nuclear immunostaining for PRAME as weak, moderate, or strong (1+, 2+, or 3+, respectively). Sebaceous glands were used as an internal control to confirm the functioning of the PRAME antibody stain. These characteristics of immunoexpression were estimated by both dermatopathologists during the review of the cases.

## 3. Results

### Features of the Examined Populations

A total of 275 melanocytic proliferations were examined, including 136 benign lesions, 12 lesions of the uncertain potential of malignancy, and 127 malignant lesions. In total, 156 (56.7%) patients were male, and 119 (43.2%) patients were female, with a mean age of 56.8 in the first group and 43.7 in the second group. Of the 136 melanocytic benign lesions, 84 (61.7%) were nevi with low-grade dysplasia (according to WHO 2018 [14]), and 52 (38.2%) were nevi with high-grade dysplasia. Of the 12 melanocytic lesions with uncertain malignancy potential, 4 were classified as melanocytic tumors of uncertain malignant potential (MELTUMP), 3 were classified as superficial atypical melanocytic proliferations of unknown significance (SAMPUS), and 5 were classified as Spitz nevi with high dysplasia degree (SPARK nevus). Finally, regarding the 127 malignant lesions, 68 (53.5%) lesions were superficial spreading type melanoma (SSM), 17 (13.3%) lesions were classified as lentigo maligna type melanoma (LMM), 7 (5.5%) as pagetoid type melanoma, and 12 (9.4%) as nodular melanoma (NM). Furthermore, 10 cases (7,8%) were categorized as spindle cell melanoma and 6 cases (4,7%%) as nevoid melanoma. Finally, the other seven cases (5,5%) were diagnosed as desmoplastic melanoma.

Of the 136 benign melanocytic lesions, divided into 84 with low-grade dysplasia and 52 with high-grade dysplasia, 81/84 (96.4%) were negative or had a score of 1+ for PRAME, while 4/52 (7,6%) nevi with high-grade dysplasia were positive for PRAME with an immunoscore of 1+/2+ (Figure 1A). Regarding the 12 lesions with uncertain potential for malignancy, 4 MELTUMP, 3 SAMPUS, and 5 SPARK nevus were characterized by heterogeneous immunostaining for PRAME with areas of greater staining alternating with areas of lower scores (3+ vs. 2+/1+) (Figure 1B). Finally, regarding the cases of MM, 61/68 cases of SSM (89.7%) were positive with an immunoscore of 4+, and the remaining 7 cases (10.2%) were positive with a value of 3+ (Figure 1C). Of the 17 cases of LMM, 16/17 (94.1%) were positive with an immunoscore of 4+, and 1 case (5.9%) was positive with 3+ (Figure 1D). Regarding the seven cases of pagetoid MM, six cases (85.7%) were positive with a value of 4+ and one case (14.2%) was positive with a value of 3+ (Figure 2A). Finally, regarding the 12 cases of NM, 11/12 cases (91.6%) expressed PRAME with a value equal to 4+ (Figure 2B). As regards, instead, the more particular histotypes of MM, of the 10 cases of spindle cell melanoma, 4 cases (40%) were positive with an immunoscore of 2+, and 6 cases (60%) were positive with an immunoscore of 3+. In comparison, of the six cases of nevoid MM, five cases (83.3%) were positive with an immunoscore of 2+, and one case (16.6%) was positive with 3+. Regarding the DM, six of the seven cases were negative, while one case was positive with an immunoscore of 1+.

## 4. Discussion

In recent years, the continuous need to refine the diagnosis of MM associated with the difficulty and inter-observer variability of atypical pigmented lesions has made it possible to search for new markers that are useful in both the diagnostic and potentially therapeutic fields [15].

Although initially discovered in tumor-reactive T-cell clones with cutaneous melanoma [16] and used in some clinical trials in therapeutic terms [17], in the last few years, PRAME has been evaluated and considered a reliable marker for the differential diagnosis of MM [18]. In this context, the paper by Lezcano et al. [18] was one of the first to investigate the usefulness of PRAME in melanocytic lesions. The authors analyzed 400 melanocytic lesions divided into 155 melanomas, 100 metastatic melanomas, and 145 pigmented melanocytic lesions. Among the data presented by the authors, 83% of MM were widely positive for PRAME, a figure very similar to that in our study in which we reported positivity for PRAME (with a score of 3+/4+) in 81.1% of MM. Among the various subtypes, we did not have cases of acral MM (as instead were observed in the study by Lezcano et al.) However, we had 89.7% positivity in the SMM, a figure very similar to the 92.5% of SMM in the paper by Lezcano et al.; furthermore, in our study, 94.1% of the LMMs were positive for PRAME with sore 4+, a value a little higher than that reported in the previous paper (88.6%), while we recorded a 91.4% positivity for PRAME in the case of NM, a value very similar to the 90% found by Lezcano et al. In our series, we also reported seven cases of pagetoid MM, of which 85.7% were positive for PRAME with a score of 4+. In relation to the less common MM histotypes, we reported 10 cases of melanoma spindle cells, with PRAME immunoscores ranging between 2+ and 3+. Furthermore, we reported six cases of nevoid melanoma with scores between 2+ and 3+. Finally, with regard to DM, we found a percentage 35% less than that observed in the previous study by Lezcano et al., with one of seven cases (14.2%) positive with an immunoscore of 1+.

Regarding the pigmented melanocytic lesions, of the 145 cases reported by Lezcano et al., 86.4% were completely negative for PRAME, with 13.6% of mild and focal positivity in some nevi such as dysplastic, recurrent, etc. In our work, we found a slightly higher negative value, equal to 96.4% for nevi with low-grade dysplasia (according to WHO 2018), and we reported a positivity rate for PRAME (7.6%) in nevi with high-grade dysplasia with a score of 1+/2+. These data are particularly interesting because they seem to confirm the trend of the results reported in the paper by Gassenmaier et al. [19] in which PRAME proved to be very reliable in the distinction between nevi with high-grade dysplasia and thin malignant melanomas (<1.00 mm). Furthermore, also in the paper by Du j et al. [20], the authors reported their experience with PRAME related to the differential diagnosis between dysplastic nevi and MM. Indeed, 26/50, 52.0% of cutaneous primary showed diffuse positive PRAME staining, and 40/40, or 100%, of common nevus and 8 (8/8) cases of dysplastic nevus were PRAME-negative. Therefore, compared with the melanocytic nevus group, the melanoma group included more cases with diffuse positive PRAME staining. These data are perfectly in agreement with the data of our study, even if we reported a percentage of only 4/52 cases of severe dysplastic nevi positive for PRAME (7.6%).

Regarding the 12 lesions with uncertain potential for malignancy comprising 4 MELTUMP, 3 SAMPUS, and 5 SPARK nevus, we also reported a certain degree of heterogeneity of immunostaining for PRAME, characterized by areas of greater positivity (score 3+) and areas of positivity equal to 2+ or also 1+. As already addressed by Raghavan et al. [21], the lesions with intermediate and/or spitzoid histopathological characteristics deserve particular attention, not only in morphological terms but also with regard to immunopositivity for PRAME. In fact, in their paper, the authors reported data very similar to ours, with 95.8% of benign melanocytic lesions completely negative for PRAME (mitotically active nevi, traumatized nevi, nevi with persistence/recurrence characteristics, and dysplastic nevi), but they differ from the results of our study, as they reported only one case of atypical Spitz tumor diffusely positive for PRAME (the other two consisted of a Spitz nevus and a spitzoid melanoma). In our work, however, we reported a higher rate of positivity in the constituent melanocytes MELTUMP, SAMPUS, and SPARK nevus, sometimes with a score of even 3+. We believe these data are very important to impose caution on the immunohistochemical interpretation of PRAME in the lesions that already have difficult nosographic classification.

## 5. Conclusions

In conclusion, our study also supports the evidence already present in the literature, attributing a definitely significant role to PRAME in the complex differential diagnosis between dysplastic melanocytic nevi and malignant melanomas, while remembering that there is a subgroup of lesions that can express, even quite intensely, this immunomarker. In the various histotypes of malignant melanoma, PRAME appears to be quite reliable, even considering that some particular types, including nevoid melanoma, spindle cell melanoma, and desmoplastic melanoma (above all), tend to lose immunopositivity, and therefore, the dermatopathologist must know this eventuality. Overall, PRAME is useful but not conclusive when applied to lesions with an uncertain potential for malignancy such as MELTUMP, SAMPUS, and SPARK nevus. Future studies with more case series and other data are needed to elucidate and better understand the usefulness of PRAME in other MM subtypes and other primary skin lesions.

## Figures and Tables

**Figure 1 diagnostics-12-02197-f001:**
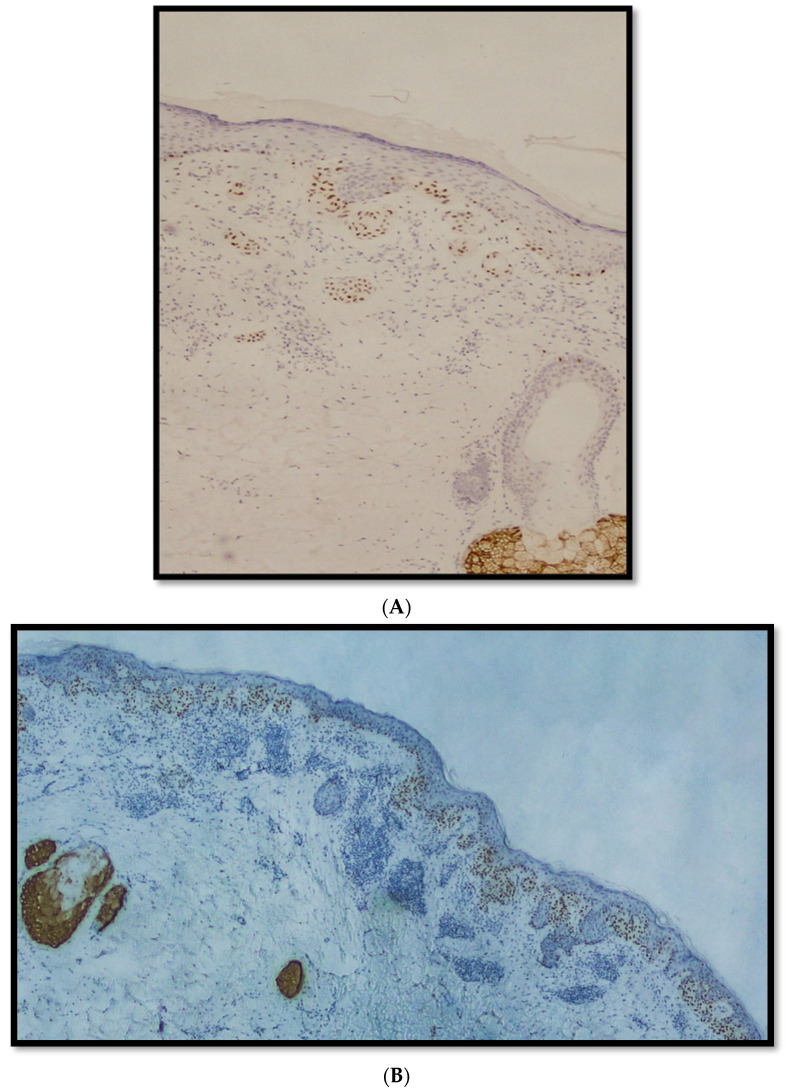
(**A**) Immunostaining for PRAME in a nevus with high-grade dysplasia sec. WHO 2018. Note some nuclear positivity in junctional melanocytes. Bottom right there is a sebaceous gland used as control (immunostaining for PRAME, original magnification 10×); (**B**) immunostaining for PRAME in a lesion classified as MELTUMP: note the heterogeneous nuclear staining of nests of melanocytes with areas of positivity and negativity (immunostaining for PRAME, original magnification 10×); (**C**) example of strong nuclear immunopositivity for PRAME in an SSM. Note the nests of neoplastic melanocytes with elongated rete ridges and a focal aspect of invasivity (red arrow); (**D**) an example of LMM immunostained with anti-PRAME antibody: note the strong nuclear positivity of neoplastic cells above the dermo-epidermal junction, with occasional pagetoid spreading and some focus of melanocytic conglomeration (immunostaining for PRAME, original magnification 10×).

**Figure 2 diagnostics-12-02197-f002:**
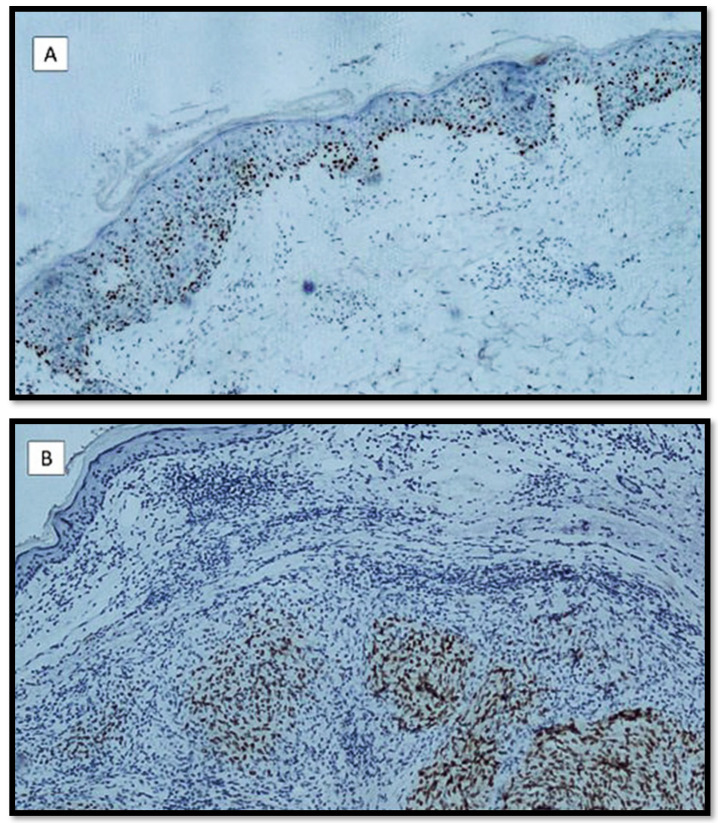
(**A**) In this micrograph, an example of pagetoid histotype of MM is shown: note the diffuse pagetoid spreading of melanocytes strongly positive for PRAME (immunostaining for PRAME, original magnification 10×); (**B**) an example of nodular melanoma: note the positivity of melanocytes within dermis without any superficial spreading component (immunostaining for PRAME, original magnification 10×).

## Data Availability

Not applicable.

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
