# Peer review of "PRAME Immunoexpression in 275 Cutaneous Melanocytic Lesions: A Double Institutional Experience"

_diagnostics, 2022, doi:10.3390/diagnostics12092197_

Round 1
Reviewer 1 Report
The reviewer wishes to thank the editor and the authors for the opportunity to review this thought provoking article. This manuscript seeks to further explore PRAME immunoperoxidase staining in a variety of melanocytic lesions including benign, intermediate, and malignant categories. The results appear in keeping with the current literature, although the intermediate category results suggest that PRAME staining should be used in only conjunction with morphology and other immunoperoxidase stains in order to determine the true nature of a lesion.
In order to help the audience best interpret the study (and indeed use the study in their daily practice), a further detailed explanation of the scoring system ( Section 2.2. Prame immunostaining, line 105-108) with exact percentages and further definitions of intensity may be of help.
Additionally the manuscript may be more attractive if figures are modified. Figures 1A, 1B, 1C, 1D, 2A and 2B all appear to be very dark or blue in color. Figures 1B, 1C, 1D, 2A, and 2B also need to be of higher resolution. Figures 1A, 1B and perhaps 2B may benefit from the addition of an H&E for comparison purposes. Figure 2B may also be more attractive if more of the overlying epidermis is seen.
Overall, the article is enlightening.
Author Response
Reviewer n’1: In order to help the audience best interpret the study (and indeed use the study in their daily practice), a further detailed explanation of the scoring system ( Section 2.2. Prame immunostaining, line 105-108) with exact percentages and further definitions of intensity may be of help.
Answer n’1: Dear Reviewer n’1, first of all, thank you very much for your wonderful words. We are very happy. Yes, we improved the detailed explanation of the scoring system. Thank you very much.
Reviewer n’1: Additionally the manuscript may be more attractive if figures are modified. Figures 1A, 1B, 1C, 1D, 2A and 2B all appear to be very dark or blue in color. Figures 1B, 1C, 1D, 2A, and 2B also need to be of higher resolution. Figures 1A, 1B and perhaps 2B may benefit from the addition of an H&E for comparison purposes. Figure 2B may also be more attractive if more of the overlying epidermis is seen.
Answer n’2: Dear Reviewer n’1, thank you. We modified figures 1A, 1B, 1C, 1D, 2A and 2B and we try to improve the resolutions. We are unable to find the H&E sections because the cases are all in archive. We hope that this will be not a problem. Thank you for all, the authors.
Reviewer 2 Report
This is a novel topic with a new technique to be used
The sample should be larger to give more significance and power
Author Response
Reviewer n'2:
This is a novel topic with a new technique to be used
The sample should be larger to give more significance and power
Answer n'1: Thank you so much. In the next work we will increase the casuistry.
Reviewer 3 Report
The manuscript is of interest and overall well-written. I would suggest the authors to address the following minor issues:
- Please report if IRB approval was obtained for the study
- Was sample size estimation performed?
- The information on the types of lesions (lines 85-97) should be moved to the results. In the results section, the authors are suggested to add a paragraph entitled sample characteristics, where data on gender/age distribution of patients and classification of the lesions should be reported.
Author Response
Reviewer n'3:
The manuscript is of interest and overall well-written. I would suggest the authors to address the following minor issues:
- Please report if IRB approval was obtained for the study
- Was sample size estimation performed?
- The information on the types of lesions (lines 85-97) should be moved to the results. In the results section, the authors are suggested to add a paragraph entitled sample characteristics, where data on gender/age distribution of patients and classification of the lesions should be reported.
Answer n'1: Dear Reviewer n'3, thank you very much for this wonderful words. We are very proud. Ok, we add the sentence about the consensus form by local ethical committee; furthermore, we have add a paragraph in "results" section with new data about gender and age. Finally, we moved the lines 85-97 into results section.
Thanks again